# The Modified versus the Conventional Winograd Technique for the Treatment of Onychocryptosis: A Retrospective Study

**DOI:** 10.3390/ijerph19137818

**Published:** 2022-06-25

**Authors:** Flávio Oliveira, Joaquín O. Izquierdo-Cases, Alfonso Martínez-Nova, Elena Contreras-Barragán, Pedro V. Munuera-Martínez

**Affiliations:** 1Escola Superior de Saúde da Cruz Vermelha Portuguesa, 1300-125 Lisbon, Portugal; investigacao.podiat@gmail.com; 2Faculty of Health Sciences, Universidad Católica San Antonio de Murcia, 30107 Murcia, Spain; doctoroscarizquierdo@gmail.com; 3Department of Nursing, Universidad de Extremadura, 10003 Caceres, Spain; podoalf@unex.es; 4Private Practice, Calle Manuel Sánchez, 21006 Huelva, Spain; elena.con.bar@gmail.com; 5Department of Podiatry, University of Seville, 41009 Seville, Spain

**Keywords:** ingrown toenail, Onychocryptosis, surgery, Winograd, foot

## Abstract

The Winograd technique is a common surgical treatment for ingrown toenails. Attempting to improve the results of this technique, two modifications were adopted: the incisional approach and the use of adhesive approximation strips. This study aimed to compare the conventional technique and the modified version based on (i) postoperative complications, (ii) healing time, (iii) recurrence rate, and (iv) overall patient satisfaction. A longitudinal, observational, and retrospective design was used, with a sample of 208 patients divided into the modified Winograd technique (n = 111) and the conventional Winograd technique (n = 97) in three clinics in Portugal, with follow-up periods of more than 15 years and 10 years, respectively. The modifications to the Winograd technique revealed fewer postsurgical complications, in terms of infections (1.8% vs. 20.62%, *p* < 0.010), recurrence rate (2.7% vs. 5.21%, *p* > 0.05), shorter recovery time (8.10 ± 0.76 vs. 14.51 ± 3.48 days, *p* < 0.001), and lower postoperative pain and better satisfaction with the functional and esthetic results, with the patient’s overall satisfaction, and with significant differences in relation to the conventional technique (*p* < 0.001). The modifications performed showed a lower rate of infection, decreased healing time, and better patient satisfaction, suggesting that it may be adopted in clinical practice for the treatment of stages II and III ingrown toenails.

## 1. Introduction

Ingrown toenail is a common and painful nail condition in which the nail plate penetrates the nail fold, causing swelling, pain, inflammation, and, in more severe cases, infection and formation of granulation tissue [1,2,3,4]. These painful symptoms can lead to an increase in morbidity and absenteeism from work [1,2].

The lateral fold of the big toe is the most often affected, although other toes can also be affected [5]. Etiologically, biomechanical changes, pathological nail plate curvatures, incorrect cutting, and type of shoes are the main causes of ingrown toenails [3,6,7].

The severity of ingrown toenails is classified into stages with a progressively increased level of inflammation of the nail fold. Kline [8] developed a classification system based on five stages of severity: I or mild, II or inflammation, III or infection, IV or infection with onycholysis, and V or infection with bilateral onycholysis.

For less severe cases, a conservative treatment can be used. A severe inflammation or infection can be solved with surgery, but an infection with onycholysis needs a resection to control the adhesion of the toenail before the surgery [8]. Although clinicians favor noninvasive methods to protect the lateral nail fold [9], some researchers have questioned the adoption of conservative and nonsurgical methods, due to the high rate of recurrence, suggesting instead surgical approaches, even if they are more invasive [10,11].

The Winograd technique is a surgical technique usually used in stages II and III of ingrown toenails [4,6,10], when the nail fold presents hypertrophy and needs to be resected [12,13,14]. Despite being a technique used since 1929 [15], few changes have been made to it, and the focus has been on the type of suture [16], the number of stitches [16], or the limit of resection [17]. Recent research on improvements to this technique has not been found in the literature, so we propose to compare the Winograd technique with a modified version based on (i) postoperative complications, (ii) healing time, (iii) recurrence rate, and (iv) overall patient satisfaction. Likewise, we intended to compare the safety and efficacy of these modifications to improve results vis-à-vis the conventional Winograd technique. 

## 2. Materials and Methods

This was a retrospective study on a series of surgical cases performed in one setting over a specific period to compare the results of two surgical techniques. It was conducted in clinics in Vila Nova de Famalicão, Viana do Castelo, and Lisbon, Portugal between February 2003 and September 2019. The patients included gave informed consent and presented with ingrown toenail stages II or III, according to Kline’s classification [8]. The patients with bone lesions, cystic lesions, connective tissue pathologies, history of pathological healing, pregnant or lactating women, and patients who did not respect the postsurgical indications were excluded. A total of 208 patients were enrolled in this study; 111 were treated with the modified Winograd technique, and 97 were treated with the conventional Winograd technique 

All procedures were approved by the Ethics Committee of the University Institute of Health Sciences—CESPU, according to the Helsinki Declaration. Patients were informed of all the details of both interventions. 

Local anesthesia of the hallux with the Frost H technique with 3% Mepivacaine Hydrochloride was administrated following surface cleaning with a chlorohexidine sponge and povidone-iodine 10%. Hemostasis was performed with an ischemic ring or *Smarch* strap, and the periungual hypertrophic soft tissue to be removed was evaluated. 

For patients treated with the conventional Winograd technique, a longitudinal incision in the eponychium 2–3 mm from the fold affected and a second incision with an elliptic form allowed to excise partially the nail plate and the matrix and also perform curettage of all the hypertrophied tissue. The fold and nail were approximated using adhesive bandages suture.

For patients treated with the modified Winograd technique, the first modification was performed at the incision. Instead of detaching the nail plate, three incisions were made: (i) a first dorsal incision with a n° 15 scalpel blade (Figure 1a,b), in a posteroanterior direction, from the posterior eponychium to the distal eponychium; (ii) a second incision made with the cutting bevel of the blade oriented dorsally (Figure 1c), in an anteroposterior direction with the first incision, in depth from the hyponychium to the cortical of the dorsal base of the distal phalanx; and (iii) a third parabola incision (Figure 1d), from the posterior eponychium to the hyponychium, this being the only incision in common with the conventional Winograd technique. After these three incisions had been made, we were able to extract the anatomical piece, consisting of a portion of nail matrix and lamina, a portion of nail bed, granulation, and/or fibrous tissue and some residual adipose tissue. After mechanical extraction, the second modification was performed by using approximation adhesive strips (Figure 1e,f). All three strips were applied at spaces of about two millimeters to allow normal drainage of these incisions, without overlapping in the plantar area of the digit so that there was no vascular constriction regarding possible edema of the toe. The strips were covered with nonsticky dressing, replaced every two, four/five days, and at seven/eight days, the patients returned to full normal activity. 

Patients in both groups were indicated to have 400 mg ibuprofene/8 h for two days. Antibiotic therapy was not prescribed prior to the surgery. 

All patients were advised to keep their feet elevated (at the level of the pelvic girdle), to walk for five minutes every hour from 12 h after intervention, to wear postoperative footwear, and to not remove or wet the dressing. From the first dressing change, in which the amount of bleeding was assessed, to the last, the evolution was always checked, verifying whether there was inflammation, infection, or other variables of interest, as described below. After total recovery, three follow-up visits were carried out, at six months, one year, and two years. 

To evaluate the treatment outcomes of both Winograd techniques, the following variables were defined: Type of bleeding: mild/moderate (partially/totally staining the cellulose dressing and partially staining the gauze in contact with it) or heavy (staining the dressing and most of the gauze, showing staining on the bandage);Postsurgical complications: infection (yes, no);Recurrences: inclusion cysts (yes, no);Healing time: when patient could return to work (average in days);

Overall patient satisfaction: postoperative pain level, functional outcome, esthetic outcome, and degree of patient satisfaction. A Visual Analog Scale (0–10) was used to assess postoperative pain level, which was considered relevant if it was ≥7. For the functional outcome, the ability to wear shoes, discomfort, performance of daily activities, and practice of physical activity were evaluated. For the esthetic result, the width and color of the nail and the appearance of the scar were evaluated. Global satisfaction analyzes the whole treatment process as well as outcomes perceived by the patient. A classification ≥9 was considered very satisfactory for these outcomes, on a scale of 0 (not satisfied) to 10 (very satisfied). These outcomes were collected with a satisfaction questionnaire that was also distributed to patients after total healing. Patients who had already completed recovery before the beginning of this research study were contacted by phone and asked to fill out the questionnaire in person or via e-mail. 

Except for mean time to total healing, a continuous variable that requires mean and standard deviation, the remaining categorical variables are reported in terms of relative and absolute frequencies. Overall patient satisfaction was quantified using cut-offs, ≥7 for pain and ≥9 for functional, esthetic, and global satisfaction. Comparisons were made using Pearson’s chi-square test for nominal variables. The Mann–Whitney U test was used for continuous and ordinal variables with non-normality (according to the results of the Kolmogorov–Smirnov test) in comparison to the independent groups. Size effect was calculated for all significant differences in the outcome of treatment variables: Hedges’ g for healing time and Phi (φ) for type of bleeding and infection. The size effect was classified according to the following for the Hedges’ g: 0.2 small, 0.5 medium, and 0.8 large. The size effect was classified according to the following for Phi: 0.1 small, 0.3 medium, and 0.5 large. All data were statistically analyzed using SPSS 25.0 software, and *p* ≤ 0.05 was considered statistically significant.

## 3. Results

A total of 208 patients with stages II or III ingrown toenails were treated with the Winograd technique, 97 with the conventional method (58 males [59.79%] and 39 females [40.21%], mean age 27.23 ± 14.02) and 111 with the modified method (45 males [40.54%] and 66 [59.46%], mean age 34.22 ± 18.63). A statistical difference was found between the two groups, in both gender (*p* = 0.004) and age (*p* = 0.010) (Table 1).

The results of the ingrown toenail treatment with the conventional and modified Winograd techniques relating to the type of bleeding and infection as postoperative complications, recurrences, and healing time are listed in Table 1. Except for recurrences, which didn’t show significant differences between the two techniques (*p* > 0.05), the remaining variables presented statistically significant differences. Abundant bleeding, which increases the risk of infection, was significantly less in the modified Winograd technique group (*p* < 0.01 for bleeding; *p* = 0.010 for the rate of infection). The time of healing was also significantly shorter in the modified Winograd technique group (*p* < 0.001).

Overall patient satisfaction was also measured, based on pain, functional and esthetic outcomes, and global satisfaction. With a cut-off of relevant pain at 7 (scale 0–10), a statistically significant difference was found between the two groups, with less postoperative pain in the modified technique group (*p* < 0.001). Considering a cut-off of 9 (scale 0–10), the functional, esthetic, and global patient satisfaction outcomes showed greater satisfaction in the modified technique group, with statistically significant differences (*p* < 0.001) (Table 2). 

## 4. Discussion

Ingrown toenail is a common and painful condition that, in more severe cases, needs to be treated with surgical procedures. Because the Winograd technique is one of the most common techniques, this study aimed to compare two approaches to this technique, a conventional one and a modified one, to evaluate the effectiveness of modifications for the treatment of ingrown toenails. 

Some authors reporting on this technique have presented recurrence rates of 12% [4,6], an average postoperative infection rate of 10% [6], and a high degree of functional and esthetic patient satisfaction, both above 90%, as verified by large-scale studies [6,18,19]. In order to decrease recurrence as well as postoperative rates and healing time, an approach was developed that involves two modifications to the originally described technique [15]. 

The analysis of the type of bleeding assesses whether it was light/moderate or heavy in the postsurgical period, considering that the most satisfactory result corresponds to the lowest possible level of bleeding because of the increased probability of developing inflammation and even infections. Although heavy bleeding was less frequent in both techniques, there are still significant differences (*p* < 0.001), since it is even more frequent when the conventional technique was used, compared to the modified one. These differences may be related to the rate of infection as a postsurgical complication, given that this rate depends on the technique used (*p* = 0.010), and it was more frequent with the conventional technique. Recent studies using the conventional Winograd technique reveal signs of infection of 7.69% in 65 patients [6], 6.89% in 23 patients [20], 3.2% in 95 patients [17], and 3.5% in 224 patients [18]. Our conventional sample presented higher results than those described; however, the modified sample was within the minimum parameters. This reduced rate of infection can also be due to the combination of the two modifications, with emphasis on the use of adhesive sutures, which reduce tissue edema around the sutures, allow more efficient drainage, and reduce the possible adhesion of the dressing. In addition to reducing the likelihood of infection, this modification may contribute to a reduced level of postsurgical pain [21].

The recurrence rate is, probably, one of the more representative indicators of the success of the treatment, and it depends on the follow-up time. With a large retrospective follow-up (15 and 10 years for modified and conventional techniques, respectively), the recurrence rates were not statistically different (*p* > 0.05), but both presented low rates (2.7% vs. 5.21%). For shorter follow-ups, such as one year, the recurrence rate varied between 13.2% in 75 interventions [22], 10.34% in 29 interventions [20], and 6% in 50 interventions [23]. With a follow-up of 15 months, 3.2% of 39 patients [17] and 9.4% of 85 patients [24] complained of recurrence. 

Postsurgical complications have a direct relationship to total healing time, and, in the same way, patients expect to return to their activities as soon as possible after surgery. In this study, a statistically significant difference between the two techniques (*p* < 0.001) was observed, with a higher healing time in the conventional technique compared to the modified technique, which allows us to conclude that the recovery time depends on the technique adopted. Although there are differences, these mean recovery times (8.10 ± 0.76 vs. 14.5 ± 3.48) are within the limits identified by other authors, with a range of 2.8 ± 1.2 days [17], 13.8 ± 2.26 days [24], 13.62 days [6], and 10 days [23] until a return to work. These mean times cannot be analyzed in isolation, as they depend on postsurgical complications, which, in addition to being a constraint on patients’ lives, greatly delay healing. Therefore, it was expected that the modified technique, which registered much fewer complications, would also have a shorter recovery time, as verified.

Overall patient satisfaction included postoperative pain, which can be highly disabling, and functional, esthetic, and global satisfaction, due to the consequences for the patient’s quality of life. Postoperative pain was significantly higher in patients submitted to the conventional Winograd technique (*p* < 0.001). Relevant complaints were identified in 0.9% and 12.37% of patients submitted to the modified and conventional techniques, respectively. Kose et al. [22] reported 2.67% of complaints of intermittent postoperative pain using the conventional Winograd technique. The low rates of postsurgical complications and recurrence, associated with the use of adhesive sutures, which reduces tissue edema, might be the main reasons for the lower pain rate presented by patients undergoing the modified technique. The pain index does not appear as one of the indicators measured in several comparative studies of the Winograd technique, but it is essential to include because the choice of a technique might have to consider the least possible pain. 

In terms of functional and esthetic outcomes and global satisfaction, there are statistical differences between the two techniques (*p* < 0.001), with better outcomes for patients submitted to the modified technique. These differences in functional outcomes might be due to the postoperative recovery time, which allows a quick reintroduction into activities of daily living, increasing the perception of functionality. In some patients, the conventional Winograd technique might cause a narrowing of the nail that can be seen as cosmetically less favorable [19], particularly by females [6,20,22]. For example, Huang et al. [17] and Ali et al. [6] showed 5.26% and 7.69% of complaints due to the esthetic result. The modified Winograd technique prevents not only this narrowing but also scars, due to the type of suture used, and this might be the reason for the difference between the two approaches (94.59% in the modified technique vs. 75.26% in the conventional technique). In general, the Winograd technique presents high levels of satisfaction, such as 97% [23], 92.6% [22], 92.30% [13], or 92% [23]. In some studies, global satisfaction is lower, 82.4% [24] and 69.56% [21], due to complaints related to the esthetic appearance and to recurrence.

This study has several potential limitations. One of them is that both groups were not similar regarding age and sex. After the operations of patients with the conventional technique years ago, the modified technique started to be used, and patients who searched for treatments for ingrown nail and joined the characteristics previously mentioned, were included as the second group. Therefore, this study retrospectively compared two versions (conventional and modified) of the same surgical technique on patients who were not randomly allocated to both groups but were attended continuously. The lack of randomization, which may be considered as another limitation, could be the cause of these differences. Different study designs, as randomized clinical trials, are needed to achieve more rigorous conclusions.

## 5. Conclusions

In conclusion, the modifications to the Winograd technique in the incision approach and type of suture presented better outcomes in terms of the infection rate, healing time, and patient satisfaction, maintaining the low rate of recurrence compared to the conventional technique. More studies are needed, particularly with questionnaires to evaluate the perspective of patient satisfaction and bleeding to support scientific evidence. However, we consider this technique safe and easy to perform in clinical practice for the treatment of stages II and III ingrown toenails.

## Figures and Tables

**Figure 1 ijerph-19-07818-f001:**
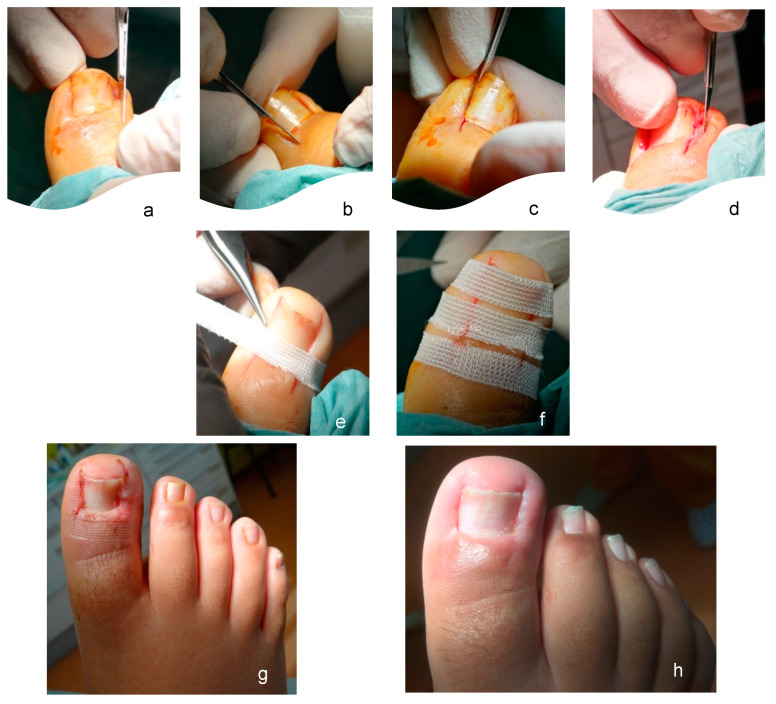
First modification of the Winograd Technique, incision in the eponychium (**a**,**b**), incision from distal to proximal of the nail plate (**c**), and incision in parabola and oblique movement (**d**). Second modification of the Winograd Technique, using adhesive sutures, with sequential and distal placement, the first strip is superimposed on the eponychium (**e**) and the last on the distal portion of the nail plate (**f**). Seven days postsurgical image (**g**) and 21 days postsurgical image (**h**).

**Table 1 ijerph-19-07818-t001:** Outcome of treatment with modified and conventional Winograd techniques comparison.

Variables	Modified Winograd Technique (n = 111)	Conventional Winograd Technique (n = 97)	*p*-Value	Size Effect
Gender			0.004	
Male	45 (40.54%)	58 (59.79%)		
Female	66 (59.46%)	39 (40.21%)		
Age	34.22 ± 18.63	27.23 ± 14.02	0.010	
Type of Bleeding			<0.001	1.34 *
Light/moderate	109 (98.20%)	77 (79.38%)		
Abundant	2 (1.80%)	20 (20.62%)		
Postoperative complications			0.010	0.36 *
Infection	5 (4.50%)	13 (13.40%)		
Recurrences			>0.05	-
Inclusion cysts	3 (2.7%)	5 (5.21%)		
Healing time	8.10 ± 0.76	14.51 ± 3.48	<0.001	2.63 **

Data presented as n (%) and mean ± standard deviation. * Phi was calculated for size effect. ** Hedges’ g was calculated for size effect.

**Table 2 ijerph-19-07818-t002:** Outcomes of overall patient satisfaction with modified and conventional Winograd techniques comparison.

Variables	Modified Winograd Technique (n = 111)	Conventional Winograd Technique (n = 97)	*p*-Value
Pain	1 (2) [0–7]	3 (5) [1–9]	<0.001
Functional outcome	10 (0) [8–10]	10 (1) [6–10]	<0.001
Esthetic outcome	10 (0) [2–10]	9 (1) [5–10]	<0.001
Global patient satisfation	10 (0) [8–10]	10 (1) [4–10]	<0.001

Data presented as median (interquartile range) and [min–max].

## Data Availability

Not applicable.

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
