# Peer review of "The Modified versus the Conventional Winograd Technique for the Treatment of Onychocryptosis: A Retrospective Study"

_ijerph, 2022, doi:10.3390/ijerph19137818_

Round 1
Reviewer 1 Report
"The modified versus the conventional Winograd technique for the treatment of onychocryptosis: a retrospective study".
English language and style are fine without significant corrections
This study, an original modified treatment technique for onychocryptosis, presents an appropriate design with a suitable method that leads to very interesting results and conclusions for the readers of this journal
Author Response
Dear Editor and Reviewer 1,
Thank you very much for your time and effort on reading this manuscript, and for your comments.
Sincerely,
The authors.
Reviewer 2 Report
The manuscript compared with Modified Winograd technique and conventional Winograd technique. The work is interesting to audiences who are worried about the ingrown nail.
Thus, I have just a few comments below.
Major
1) Did the authors search the cause of ingrown nail? This is because you searched the recurrence rate. Ingrown nail is related to foot deformity.
2) Could the authors explain why the authors select the Winograd technique instead of phenolization, NaOH and labiomatricectomy? Depending on the area, surgeon prefer other method.
3) Did the patients use painkiller in post operation term? If they used the painkiller, please include enough information about pain.
4) Age and gender have the statistical difference in this study, this needs to be addressed as well.
5) Add a picture about result of Winograd technique in figure. You need to clarify deference between modified and conventional method.
Minor
1) The sub-numberings of figures were several times forgotten in figure1.
Author Response
Authors want to thank to the reviewer for the time and effort employed to make recommendations in order to improve the manuscript. Please find below the response to your comments, and find attached the manuscript with changes as a result of your comments yellow highlighted:
Major
1) Did the authors search the cause of ingrown nail? This is because you searched the recurrence rate. Ingrown nail is related to foot deformity.
AUTHORS RESPONSE: The cause of ingrown nail in the participants of this study was not recorded. A statement about cause of ingrown nail was included in the Introduction section, lines 41-42.
2) Could the authors explain why the authors select the Winograd technique instead of phenolization, NaOH and labiomatricectomy? Depending on the area, surgeon prefer other method.
AUTHORS RESPONSE: The Winograd is a surgical technique usually used in stages II and III of ingrown toenails according to Kline's classification, as stated by some authors (Ince et al, 2015; Ali et al, 2013). Also, it is used when the nail fold presents hypertrophy that needs to be resected. The participants who were operated for this study joined these characteristics. This statement was included in the Introduction section, lines 53-54.
3) Did the patients use painkiller in post operation term? If they used the painkiller, please include enough information about pain.
AUTHORS RESPONSE: Post-surgical indications were provided to all the participants of both groups. It has been included in the text.
4) Age and gender have the statistical difference in this study, this needs to be addressed as well.
AUTHORS RESPONSE: Patients who had already been operated with the conventional technique years ago were included in the study. After that, the modified technique started to be used and patients who searched for treatment for ingrown nail, and joined the characteristics previously mentioned, and completed recovery before the beginning of this research study, were included as the second group. Therefore, this study retrospectively compared two procedures (conventional and modified) of the same surgical technique on patients that were not randomly allocated to both groups but were attended continuously. The lack of randomization may be the cause of these differences. This has been addressed as a limitation in the Discussion section.
5) Add a picture about result of Winograd technique in figure. You need to clarify deference between modified and conventional method.
AUTHORS RESPONSE: The original technique is described in Materials and Methods section, lines 79-86. The modifications studied in the modified technique were two: the first one at incision (Instead of detaching the nail plate, three incisions were made), and the second modification was made by using approximation adhesive strip (Instead of the fold and nail being approximated using adhesive bandages suture). This is clarified in lines 84 to 100 in Materials and Methods section.
Two new images have been added in figure 1 with results at 7 and 21 days.
Minor
1) The sub-numberings of figures were several times forgotten in figure1.
AUTHORS RESPONSE: The missing sub-numberings in figure 1 have been included.

Reviewer 3 Report
Thanks to the authors for allowing me to review their manuscript. I attached some comments in order to improve the manuscript.
-Support the paragraph with the references alluded to (lines 57-58-59 in the introduction): Recent studies with improvements of this technique are limited however, we propose to compare the Winograd technique with a modified version based on (i) postoperative complications, (ii) healing time, (iii) recurrence rate, and (iv) overall patient satisfaction.
-I leave the following aspect to the editor for consideration, but perhaps establishing sub-sections would improve the comprehension of materials and method (e.g. design, study population, ethical aspects, description of surgical techniques, variables, statistical analysis...).
-The design of the study is not entirely clear, since not all information on the type of sampling and recruitment system is given. The study compares two different surgical techniques in the same population (the study population is well defined with adequate inclusion and exclusion criteria). But it should be better explained whether the study design involves a pre-assignment of patients with respect to surgical technique, or whether what is presented is a retrospective series of surgical cases performed in one setting over a specific period of time, where the aim is simply to compare the results of two surgical techniques. This type of study is frequent in centres where different surgical approaches to the same problem are performed (e.g. centres where there are several surgical teams made up of different professionals). In the case of the later, it is accepted that no sample size has been estimated. This is very important to consider the internal-external validity of the study. In this case, it is better to change the verb "to demonstrate" for another verb such as "to compare" when referring to the objective of the study.
-Generally, postoperative medication (analgesia) is prescribed in these procedures. If so, this should be noted, as this may influence one outcome variable noted (postoperative pain level). If in some cases antibiotic therapy was prescribed prior to surgery this should also be stated (if possible provide the number of cases). In any case, it should be explained whether these guidelines were the same or different in both groups.
Indicate which normality test was performed (Kolmogorov??) to support the use of a non-parametric test (Mann Whitney U) versus the reference test (t Student).
-I make the following points in order to improve the manuscript (it does not invalidate the publication of the study). Effect sizes can be calculated for the association of healing time (can use the g hedges or d cohen). this could have been done with the pain scores and the rest of cuantitives variables...... although I understand that perhaps the normality analysis indicated that it was better to express the results in medians and interquartile range (as opposed to means/standard deviations).
-The tables should be annotated to indicate which statistical test was used to calculate p for each association.
-The discussion is adequate and the results are well contrasted. However, no limitations in the study (e.g. inherent in the study design if it is a retrospective case series) are noted.
I congratulate the authors and encourage them to make these minor changes
Author Response
Authors want to thank to the reviewer for the time and effort employed to make recommendations in order to improve the manuscript. Please find below the response to your comments, and find attached the manuscript with changes as a result of your suggestions green highlighted:
-Support the paragraph with the references alluded to (lines 57-58-59 in the introduction): Recent studies with improvements of this technique are limited however, we propose to compare the Winograd technique with a modified version based on (i) postoperative complications, (ii) healing time, (iii) recurrence rate, and (iv) overall patient satisfaction.
AUTHOR RESPONSE: We really meant that did not find recent research on improvement of this technique, so maybe references are no required. We have changed the sentence by “Recent research on improvement of this technique has not been found in the literature, so we propose to compare the Winograd technique with […]”
-I leave the following aspect to the editor for consideration, but perhaps establishing sub-sections would improve the comprehension of materials and method (e.g. design, study population, ethical aspects, description of surgical techniques, variables, statistical analysis...).
AUTHOR RESPONSE: The Journal encouraged to use its Microsoft Word Template to submit the manuscript. This template did not show the option of sub-sections in the Method section. This template was used by the authors, who followed the instructions showed in it. Nevertheless, should the Editor indicate that sub-headings must be used we will be pleased to include them.
-The design of the study is not entirely clear, since not all information on the type of sampling and recruitment system is given. The study compares two different surgical techniques in the same population (the study population is well defined with adequate inclusion and exclusion criteria). But it should be better explained whether the study design involves a pre-assignment of patients with respect to surgical technique, or whether what is presented is a retrospective series of surgical cases performed in one setting over a specific period of time, where the aim is simply to compare the results of two surgical techniques. This type of study is frequent in centres where different surgical approaches to the same problem are performed (e.g. centres where there are several surgical teams made up of different professionals). In the case of the later, it is accepted that no sample size has been estimated. This is very important to consider the internal-external validity of the study. In this case, it is better to change the verb "to demonstrate" for another verb such as "to compare" when referring to the objective of the study.
AUTHOR RESPONSE: Certainly, it was a retrospective series of surgical cases performed in one setting over a specific period of time to compare the results of two surgical techniques. This issue has been clarified in the Method section and the verb “demonstrate” has been replaced by “compare”.
-Generally, postoperative medication (analgesia) is prescribed in these procedures. If so, this should be noted, as this may influence one outcome variable noted (postoperative pain level). If in some cases antibiotic therapy was prescribed prior to surgery this should also be stated (if possible provide the number of cases). In any case, it should be explained whether these guidelines were the same or different in both groups.
AUTHOR RESPONSE: Post-surgical instructions were the same for patients in both groups. Antibiotic therapy was not prescribed prior to surgery. Both issues have been included in the Method section.
Indicate which normality test was performed (Kolmogorov??) to support the use of a non-parametric test (Mann Whitney U) versus the reference test (t Student).
AUTHOR RESPONSE: Kolmogorov-Smirnov normality test was performed. It has been clarified in the Method section.
-I make the following points in order to improve the manuscript (it does not invalidate the publication of the study). Effect sizes can be calculated for the association of healing time (can use the g hedges or d cohen). this could have been done with the pain scores and the rest of cuantitives variables...... although I understand that perhaps the normality analysis indicated that it was better to express the results in medians and interquartile range (as opposed to means/standard deviations).
AUTHOR RESPONSE: Effect sizes for the outcome of treatment variables that were significantly different between both groups were calculated and the results have been included in table 1.
-The tables should be annotated to indicate which statistical test was used to calculate p for each association.
AUTHOR RESPONSE: The statistical tests employed are explained at the end of the Method section, lines 159 to 162.
-The discussion is adequate and the results are well contrasted. However, no limitations in the study (e.g. inherent in the study design if it is a retrospective case series) are noted.
AUTHOR RESPONSE: A limitation section stating that inherent to the study design has been included.
